# The Dynamic Risk of COVID-19-Related Events in Vaccinated Healthcare Workers (HCWs) from a Tertiary Hospital in Bucharest, Romania: A Study Based on Active Surveillance Data

**DOI:** 10.3390/vaccines12020182

**Published:** 2024-02-11

**Authors:** Carmen-Daniela Chivu, Maria-Dorina Crăciun, Daniela Pițigoi, Victoria Aramă, Monica Luminița Luminos, Gheorghiță Jugulete, Ciprian Constantin, Cătălin Gabriel Apostolescu, Adrian Streinu Cercel

**Affiliations:** 1Department of Epidemiology 1, Carol Davila University of Medicine and Pharmacy, 050474 Bucharest, Romania; carmen-daniela.chivu@drd.umfcd.ro (C.-D.C.); daniela.pitigoi@umfcd.ro (D.P.); 2Emergency Clinical Hospital for Children “Grigore Alexandrescu”, 011743 Bucharest, Romania; 3National Institute for Infectious Diseases “Prof. Dr. Matei Balș”, 021105 Bucharest, Romania; victoria.arama@umfcd.ro (V.A.); luminita.luminos@umfcd.ro (M.L.L.); gheorghita.jugulete@umfcd.ro (G.J.); catalin-gabriel.apostolescu@drd.umfcd.ro (C.G.A.); adrian.streinucercel@umfcd.ro (A.S.C.); 4Department of Infectious Diseases 1, Carol Davila University of Medicine and Pharmacy, 020021 Bucharest, Romania; 5Department of Infectious Diseases 3, Carol Davila University of Medicine and Pharmacy, 020021 Bucharest, Romania; 6Department of Clinical Sciences, Faculty of Medicine, Titu Maiorescu University, 031593 Bucharest, Romania; ciprianconstantin@metabolism.ro; 7Carol Davila Central Military Emergency University Hospital, 010825 Bucharest, Romania

**Keywords:** COVID-19, healthcare workers, vaccines protection, clinical outcome

## Abstract

Our study describes the frequency and severity of COVID-19 in HCWs and estimates the dynamic risk of COVID-19-related events. We actively surveyed all HCWs from a tertiary infectious disease hospital from 26 February 2020 to 31 May 2023. Of 1220 HCWs, 62.9% (767) had at least one COVID-19 episode. The under 29 years (*p* = 0.0001) and 40–49 years (*p* = 0.01) age groups, nurses (*p* = 0.0001), and high-risk departments (*p* = 0.037) were characteristics significantly more frequent in HCWs with COVID-19 history. A higher percentage of boosters (53.2%; *p* < 0.0001) were registered in the uninfected group. The second episode of COVID-19 was significantly milder than the first. Data regarding clinical outcomes from 31 January 2021 to 31 May 2023 were analyzed in a follow-up study to determine the risk of COVID-19-related events. The Cox regression analysis revealed that HCWs with booster shots had a lower risk of COVID-19 across all events, symptomatic events, and moderate to severe events as adjusted hazard ratio (aHR) were: 0.71 (95%CI: 0.54–0.96), 0.23 (95%CI: 0.12–0.46), and 0.17 (95%CI: 0.07–0.43), respectively. Within the vaccinated subgroup, the HCWs with hybrid immunity and booster had aHR for all followed-up events of 0.42 (95%CI: 0.30–0.58), for symptomatic events of 0.52 (95%CI: 0.36–0.74), and 0.15 (95%CI: 0.03–0.66) for moderate to severe events. The risk of COVID-19 clinical events was lower for HCWs with at least one booster than those completely vaccinated.

## 1. Introduction

The COVID-19 pandemic has impacted healthcare workers as well, leading to increased rates of morbidity and mortality. On 5 May 2023, the WHO (World Health Organization) declared COVID-19 a stable public health problem [1]. The population’s immunity acquired either naturally through infection, artificially through vaccination, or hybrid immunity and the development of antiviral drugs have all contributed to the reduction of the risks to human health.

Starting from 21 December 2020 and up to 25 October 2023, eight commercial vaccine products were used for active immunization and boosters. The WHO recommended vaccination schedules with one or two doses depending on the product. After 14 days from the primary schedule, the person is considered completely vaccinated. Homologous or heterologous boosters are also recommended 6–12 months after completing the primary schedule [2,3]. In the context of the SARS-CoV-2 Omicron variant circulation, the WHO recommends vaccination and a booster for high-risk categories, HCWs included [4].

According to the WHO, as of 25 October 2023, over 5 billion complete vaccination regimens have been administered worldwide [5]. By the beginning of 2023 in Romania, almost 8 million people were completely vaccinated, and over 2.6 million people received boosters [6]. In Romania, the adult age group distribution by vaccine regimen was as follows: in the 25–49 age group, 51.4% were completely vaccinated; 9.5% were with one booster, and only 0.1% were vaccinated with the second booster; in the 50–59 age group, 56% were completely vaccinated, 13.5% were with one booster and 0.2% were with the second booster [7]. Healthcare workers were prioritized for vaccination.

Although studies that use specific populations cannot be generalized to the entire population, they have the advantage of highlighting particular aspects. The WHO published a cohort study guide for the measuring of vaccine effectiveness among HCWs, which detailed the key aspects of the study design and survival analysis [8]. Interim real-world results on vaccine effectiveness in HCWs’ groups have been published, but longer-term data (including results in subgroups at high-risk exposure) are needed [9,10].

Our study aims to describe the frequency and severity of COVID-19 in HCWs and estimate the dynamic risk of COVID-19-related events (any SARS-CoV-2 infection regardless of the clinical outcome, symptomatic SARS-CoV-2 infection, and moderate to severe SARS-CoV-2 infection) in vaccinated and nonvaccinated subgroups. The current study continues and complements previous research on the risk of COVID-19 in HCWs in a designated hospital for treating COVID-19 patients in Bucharest, Romania [11].

## 2. Materials and Methods

### 2.1. Study Design and Study Population

This cohort study was performed at the National Institute for Infectious Diseases “Professor Dr. Matei Balș” (INBI) in Bucharest, Romania. The hospital is a reference center for managing public health alerts in Romania, including COVID-19. The first case of COVID-19 in Romania was admitted and treated on 26 February 2020 in this hospital. From April 2020 to 31 May 2023, the hospital has only treated patients with COVID-19 [12,13].

According to the type of HCW’s exposure to COVID-19 patients and their environment (direct or indirect), the hospital’s departments were classified as high-risk (emergency room, clinical, and intensive care departments) and low-risk (administrative, research, and pharmacy).

Non-pharmaceutical interventions (personal protective equipment and training) and on-site vaccination (when the vaccine became available) were employed for the protection of the HCWs. According to hospital policy, clinical evaluation and treatment of COVID-19 in HCWs were provided within the hospital. The HCWs were actively and continuously surveyed by the IPC (Infection Prevention and Control) team from 26 February 2020 to 31 May 2023. The detection of COVID-19 in HCWs was conducted through daily triage of symptoms and testing according to the national methodology and the hospital’s internal protocol with RT-PCR (reverse transcription–polymerase chain reaction) or rapid antigen test (RAT) [14]. Asymptomatic HCWs were also tested in case of direct contact with a COVID-19 confirmed case or in case of community exposures (e.g., returning from holiday or travel). COVID-19 cases were evaluated by the hospital’s infectious doctors and, depending on the severity, were either treated as outpatients or admitted to the hospital wards. A standard investigation form for COVID-19 was completed for each case and vaccination history was assessed.

The COVID-19 vaccination campaign began on 27 December 2020 and a vaccination center was established at the hospital level. COVID-19 vaccination records were computed in the National Electronic Registry of Vaccinations [15].

The HCWs population studied had 2 inclusion criteria: the employment contract within INBI “Prof. Dr. Matei Balș” and being actively on duty at least 12 months after the start of the vaccination campaign. HCWs were excluded if the professional role, the history, and the clinical outcome of the SARS-CoV-2 infection were undetermined. The case definitions according to the national methodology were applied, considering a new infection when a positive test occurred 90 days after a previous infection [14]. The HCWs were diagnosed with COVID-19 once, twice, or three times. The HCWs with three SARS-CoV-2 infections did not meet the above inclusion criteria. This resulted in a database of uninfected HCWs with one or two COVID-19 episodes. Our study population was singly censored through right censoring. The study workflow and the timeline of the surveyed events are represented in Figure 1.

### 2.2. Study Period

The HCWs were actively and continuously surveyed by the IPC team of the hospital from 26 February 2020 to 31 May 2023 (38 months). For evaluating the risk of COVID-19 in groups defined by vaccination status and COVID-19, we used the surveillance data for our follow-up study starting on 31 January 2021, when the cohort of completely vaccinated started to build up, until 31 May 2023, resulting in a 28-month (850 days) study period.

### 2.3. Data Sources and Collection

The IPC team provided the primary data source as a prospectively constituted register. The data characteristics such as age, gender, professional role, and department of activity. Vaccination status was collected retrospectively from the National Electronic Registry of Vaccinations, along with data on the date of vaccination and type of vaccine. People who received two doses of the Comirnaty, Spikevax, and Vaxzevria vaccines and those who received one dose of the Jcovden vaccine were considered completely vaccinated after 14 days from the last dose. The “Vaccinated and boosters” category was defined as HCWs who had received an additional dose of vaccine at least six months after receiving the complete vaccination schedule. “Hybrid immunity” was defined as HCWs with a laboratory-confirmed history of COVID-19 and complete vaccination. In the “Hybrid immunity and boosters” category were those HCWs with both complete vaccination schedules and boosters, as well as a history of COVID-19 during the study period.

Data related to COVID-19 exposure were extracted from the standard surveillance forms. The hospital‘s informatics (IT) system was the data source for clinical outcomes, hospitalization, and admission to the ICU. Subsequent hospitalizations of COVID-19 cases were also checked to capture worsening of the disease or complications.

In our follow-up study, we defined three COVID-19-related events: any SARS-CoV-2 infection regardless of the clinical outcome, symptomatic SARS-CoV-2 infection, and moderate to severe SARS-CoV-2 infection. Infectious disease specialists evaluated HCWs using the WHO COVID-19 clinical spectrum, considering criteria like dyspnea, chest imaging, oxygen saturation, the ratio of arterial pressure of oxygen to fraction of inspired oxygen, respiratory rate, respiratory failure, septic shock, and multiple organ dysfunction. The symptomatic individuals without dyspnea or abnormal chest imaging were classified as mild illnesses, moderate illnesses were classified as those with lower respiratory tract infection and oxygen saturation less than 94% in room air. Cases with oxygen saturation less than 94%, the ratio of arterial pressure of oxygen to fraction of inspired oxygen less than 300 mm Hg, respiratory rate over 30 breaths/min, or lung infiltrates over 50% were classified as severe [16]. Deaths were categorized as severe cases.

The resulting multiple sets represented by the register of COVID-19 in HCWs and epidemiological investigation forms, provided by the IPC team, and reports regarding COVID-19 clinical outcomes from the hospital’s IT system have been merged using a Python script and also validated manually by a human operator.

### 2.4. Statistical Analysis

Laboratory-confirmed cases of COVID-19 in HCWs were compared with non-COVID-19 HCWs based on characteristics like age, gender, professional role, department, vaccination status, type of vaccine, and booster. The first and second COVID-19 episodes were also compared based on clinical outcomes and hospitalization.

Continuous and categorical variables were presented in mean, median, interquartile ranges (IQR), numbers, and percentages. Differences between proportions of categorical variables were assessed using χ2 tests in MedCalc (MedCalc Software bv, Ostend, Belgium; https://www.medcalc.org; accessed on 25 October 2023).

We compared groups constructed based on the history of COVID-19 at the beginning of the follow-up study and the vaccination status: nonvaccinated, completely vaccinated, completely vaccinated and booster, hybrid immunity, and hybrid immunity and booster.

The Cox regression model was used, and hazard ratios (HR) were obtained. Covariates were selected from surveillance data based on the statistical significance of differences between proportions. In the nonvaccinated group, time-to-event data were the amount of time expressed in days elapsed between 31 January 2021 and the defined events. In the complete vaccinated and vaccinated and booster groups, the time-to-event data were the time elapsed 14 days after the last vaccine shot to one of the defined events. Results were expressed as adjusted hazard ratio (aHR) with 95% confidence intervals (95%CI). When the proportionality assumption was not met, the extended Cox model with time-dependent covariates was used. The threshold for significance was a *p*-value< 0.05. In addition, visualization of decreasing protection against COVID-19-related events (all events, symptomatic events, moderate to severe COVID-19) was made through Kaplan–Meier survival curves. The log-rank test was also used to compare the equality of the survival curves [17]. The statistical analysis IBM SPSS for Macintosh, version 29.0.0.0 (Armrok, NY, USA).

### 2.5. Ethics Approval

The Institutional Bioethics Committee approved the study protocol with the registration numbers C02648/16.03.2022 and C08784/27.07.2023. Due to the study’s retrospective nature, only standard informed consent was required from the participants. Access to data was limited to the IPC team and study investigators, ensuring confidentiality.

## 3. Results

During the study period, 1402 hospital employees were actively surveyed upon SARS-CoV-2 infections. Of these, 1220 were eligible for our study.

### 3.1. Overall Characteristics of SARS-CoV-2 Infected and Uninfected HCWs

The characteristics of the study population are summarized in Table 1. The majority of HCWs were female (1020/83.6%). The median age was 43.1 years, with IQR 34.2–50.4. In terms of professional roles, 37.6% were nurses, 21.1% were physicians, 24.4% were healthcare auxiliary staff, and 16.9% were nonclinical workers. A total of 69.2% of the study population worked in high-risk departments. In terms of vaccination status, 1098 HCWs (90%) were completely vaccinated with the primary schedule, and 524 HCWs (43.0%) received at least one booster. The most frequently received vaccine product was Comirnaty (1031, 93.9%). Other vaccine products were Jcovden, Spikevax, and Vaxzevria. The boosters were, in most cases (516, 98.5%), homologous as presented in Table 2.

During the 38-month active surveillance period, there were 767 (62.9%) laboratory-confirmed first episodes of COVID-19, out of which 221 (28.9%) had a second episode. A total of 453 (37.1%) HCWs remain uninfected. Those less than 29 years old were less likely to be infected (90/11.9% vs. 90/19.9%, *p* = 0.0001). Nurses were more affected than all other categories (320/41.7%, *p* = 0.0001) and those working in high-risk departments (547/71.3%, *p* = 0.03) as well (Table 1.)

Regarding the complete vaccination status, the data showed no significant differences between the first COVID-19 episode group and the uninfected group. Still, the boosters were more frequent (241/53.2%, *p* < 0.0001) in the uninfected group, as presented in Table 2.

### 3.2. Clinical Outcomes of the First and Second COVID-19 Episodes

The clinical outcome varied from asymptomatic to severe forms. Comparing the first and second episodes of COVID-19, we found statistically significant differences: the second COVID-19 episode was less severe, and HCWs were often treated as outpatients (182/82.4% vs. 457/59.6%, *p* < 0.0001). Two people died from complications of chronic diseases during the study period, and they were classified by the clinician as COVID-19 severe cases. The length of stay was significantly shorter for hospitalized patients in the second COVID-19 episode (Table 3).

At the beginning of the follow-up study, on 31 January 2021, there were 345 (28.3%) HCWs with a history of COVID-19 and 875 (71.7%) HCWs without a history of COVID-19. Applying the definitions from the study protocol resulted in the following groups: completely vaccinated and vaccinated and boosters groups without a history of COVID-19 with 405 (33.2%) and 378 (31%) subjects, respectively, and completely vaccinated and vaccinated and boosters groups with a history of COVID-19 representing hybrid immunity group with 195 (16%) subjects, and the hybrid immunity and booster group with 120 (9.8%) subjects. A total of 122 (10%) HCWs were unvaccinated.

### 3.3. The Risk of COVID-19 Events in the Follow-Up Study Population

After testing the proportionality assumption (Appendix A), we used the Cox model for moderate to severe COVID-19 events and the Cox model, with time-dependent covariates for all COVID-19 events and symptomatic events.

The analysis revealed no significant interaction between completely vaccinated HCWs and nonvaccinated ones for all of the three events evaluated but significant interaction between the nonvaccinated HCWs and the HCWs with boosters for all three events. As can be noted in Table 4 below for the vaccinated and booster group, aHR was 0.71 (95%CI: 0.54–0.96) for all COVID-19 events regardless of clinical characteristics, 0.23 (95%CI: 0.12–0.46) for symptomatic COVID-19 events and 0.17 (95%CI: 0.07–0.43). Figure 2A shows the descendent trend of the survival curves in moderate to severe COVID-19 events that indicate the decline of protection in all nonvaccinated and vaccinated groups, with higher protection of vaccinated and boosted HCWs.

### 3.4. The Risk of COVID-19 Events in Vaccinated HCWs Subgroup

Cox regression with time-dependent covariates showed significant statistical differences between all groups of vaccinated HCWs. The reference complete vaccinated group seems to have the highest risk for all three studied COVID-19-related events as all recorded proportional ratios were below one. In the hybrid immunity group, aHR for COVID-related events was 0.60 (95%CI: 0.47–0.77), 0.76 (95%CI: 0.58–0.99), and 0.40 (95%CI: 0.16–0.99), with confidence intervals very close to 1. The hybrid immunity and booster group had the lowest risk for COVID-19-related events regardless of severity, symptomatic, and moderate to severe events, as HR was 0.42 (95%CI: 0.30–0.68) and 0.15 (95%CI: 0.03–0.66). The detailed data are presented in Table 5. The proportionality assumption was also assessed (Appendix A).

The Kaplan–Meier survival curves are presented for all COVID-19 events, symptomatic events, and moderate to severe events in Figure 2B–D below, indicating a continuous decline in survival in all groups, with lower survival in completely vaccinated groups regardless of COVID-19 history. Higher protection was observed in the vaccinated and booster group, regardless of previous SARS-CoV-2 infection, but this also declined in time.

## 4. Discussion

The current study, based on active surveillance data of HCWs, evaluated risk and vaccine protection against first or second COVID-19 episodes. We obtained results regarding the frequency and severity of COVID-19 in HCWs and compared them to other studies based on real-world data. Our findings were concordant with most of the published studies. The added value of our results to already known data regarding protection provided by vaccines consists of the evaluation of this protection against any clinical outcome of COVID-19 during a longer period of time completed through active surveillance. The follow-up period was 28 months.

In our study group, 767 HCWs (62.9%) had experienced at least one episode of COVID-19. The frequency of infection was found to be higher among the 40–49 years age group, those working as nurses, and those in high-risk departments. The uninfected group had a higher percentage of individuals who had received booster shots. The clinical outcomes of the second episode of COVID-19 were milder than the first and required fewer hospitalization days as described in other studies [18,19,20,21].

The current study found that HCWs adhered in a large proportion to the primary schedule of COVID-19 vaccination. However, the low number of people who have received the COVID-19 booster doses indicates hesitancy to boosters among HCWs. This lack of confidence in vaccination is concerning and we believe that education and motivation are necessary to increase adherence to current vaccine recommendations regarding HCWs [7].

The study shows that protection against COVID-19 events after vaccination diminishes over time for all categories of HCWs, whether or not vaccinated, boosted, or with naturally acquired immunity. However, the vaccinated and booster groups have an advantage regarding protection. During the study period, the HCWs used a combination of non-pharmacological and pharmacological methods to protect themselves against COVID-19. Since the vaccinated group was well represented within the entire group, we conducted a separate follow-up risk analysis for the group of vaccinated HCWs, considering their history of COVID-19. The analysis showed that booster shots significantly reduced the risk of all three studied COVID-19 events.

A systematic review based on studies published from January to August 2021 with eight studies based on surveillance of HCW populations reported the effectiveness of the complete vaccination with Comirnaty, Moderna, and Vaxzevria vaccines of a maximum of 86% [9,10,22,23,24,25,26,27,28]. Our study showed the lowest risk of COVID-19 events in the vaccinated and booster groups and during a longer follow-up period. The SARS-CoV-2 Immunity and Reinfection Evaluation study (SIREN), a multicenter prospective cohort study of healthcare workers and support staff in the National Health Service in the UK, demonstrated the high short-term protection against COVID-19 after primary schedule and the effect of previous SARS-CoV-2 infection on protection against COVID-19 [29]. Another study on vaccine effectiveness in HCWs from the USA showed the decreasing effect of protection over time [30]. Both studies were performed close to the debut of the vaccination campaign and evaluated COVID-19-related events in a shorter period of time.

A longer follow-up period was described in a systematic review based on studies performed on the general population. This study showed decreasing effectiveness of the complete vaccination at six months follow-up but still high protection against severe clinical outcomes [31].

A study of the effectiveness of COVID-19 vaccines and duration of protection in HCWs carried out in another hospital in Romania showed the highest vaccine effectiveness for revaccinated HCWs, but the period under study was only three months. A better protection against reinfection was found in the hybrid immunity and boosters’ group [32].

The protective effect of previous COVID-19 infection was evaluated in a study in Qatar and the results showed modest protection when the previous infection had been caused by a pre-Omicron variant but strong protection when caused by a post-Omicron subvariant [33].

An explanation for the decreased protection of vaccinated HCWs, up until there are no differences between completely vaccinated and nonvaccinated, can be given by the evolving circulating variants of SARS-CoV-2. In terms of the circulation of SARS-CoV-2 variants, our study captures the periods of circulation of Alpha (March 2021–July 2021), Delta (July 2021–January 2022), and Omicron variants (January 2022–May 2023) as reported by the Romanian National Health Institute [34].

A study in Italy showed that reinfection was more frequent during the circulation period of the Omicron variant, though it featured mild or no symptoms [35]. As within our study, this study underlines the importance of boosters, as the group with the strongest protection was the one with the second boosters. In our study, there was very low coverage of the second and the third boosters in order to evaluate and conclude the importance of these interventions. The data we have obtained suggests that we need to focus our future studies on the protection of vaccines and boosters, taking into consideration SARS-CoV-2 circulating variants.

As strengths of the current study, we mention the accuracy of data records and the low risk of main biases of the study. The hospital setting is a front-line infectious disease hospital in Bucharest, Romania, where all COVID-19 cases of HCWs were managed according to European and international recommendations. We consider the risk of underdiagnosed COVID-19 cases or misclassification of the clinical outcomes to be very low, as the triage and clinical evaluation were performed by medical staff involved in front-line activities. The infection status and clinical outcome were documented and verified from multiple datasets: the surveillance register and the hospital’s IT system. Updates regarding COVID-19 clinical outcomes were recorded during the convalescent period for a rigorous classification according to WHO criteria, but long COVID-19 was not evaluated.

The vaccination status for each HCW was verified in the National Electronic Registry. The surveillance period was broad compared with other reference studies, covering the entire period in which COVID-19 was designated as a public health emergency of international concern by the WHO.

As limitations, we mention that the study evaluates only one booster, as the number of people vaccinated with multiple boosters was very low. Comorbidities were not assessed, and vaccine uptake does not take contraindications to vaccination into consideration. However, given that vaccination coverage for the initial schedule is high, and most absolute contraindications are chronic or obvious conditions at the first vaccination (e.g., anaphylaxis), the number of people with contraindications to vaccination is estimated to be low. In our medical staff structure, we had a high share of women. However, a previous study demonstrated the statistical insignificance of immune responses between genders [36]. The number of multiple COVID-19 events was too low to be analyzed, and when multiple events occurred during the follow-up, we considered only the first COVID-19 event. The hospital setting treated only COVID-19 patients during the period under study. The use of the specific COVID-19 personal protective equipment in medical activities must be added to the protection of the vaccination intervention, and the results must be interpreted in this specific context.

The study continues previous research on risk factors and the protective effect of vaccination in HCWs. Future research directions include aspects related to COVID-19 events and variants of concern (VOC) of SARS-CoV-2. In the context of the still evolving SARS-CoV-2 variant and the broadening of the spectrum of infectious diseases treated in the hospital, keeping both pharmacological and non-pharmacological measures implemented is the key to protecting HCWs.

## 5. Conclusions

The study reports the first and second episodes of COVID-19 in HCWs from a reference center of infectious disease from Romania, identifying a lower severity in the second COVID-19 episode. The risk of COVID-19-related events tended to be lower in the completely vaccinated and the vaccinated and booster groups compared to the nonvaccinated ones. HCWs with booster shots had the lowest risk for all COVID-19-related events. These findings reveal the importance of boosters in protecting HCWs against COVID-19.

## Figures and Tables

**Figure 1 vaccines-12-00182-f001:**
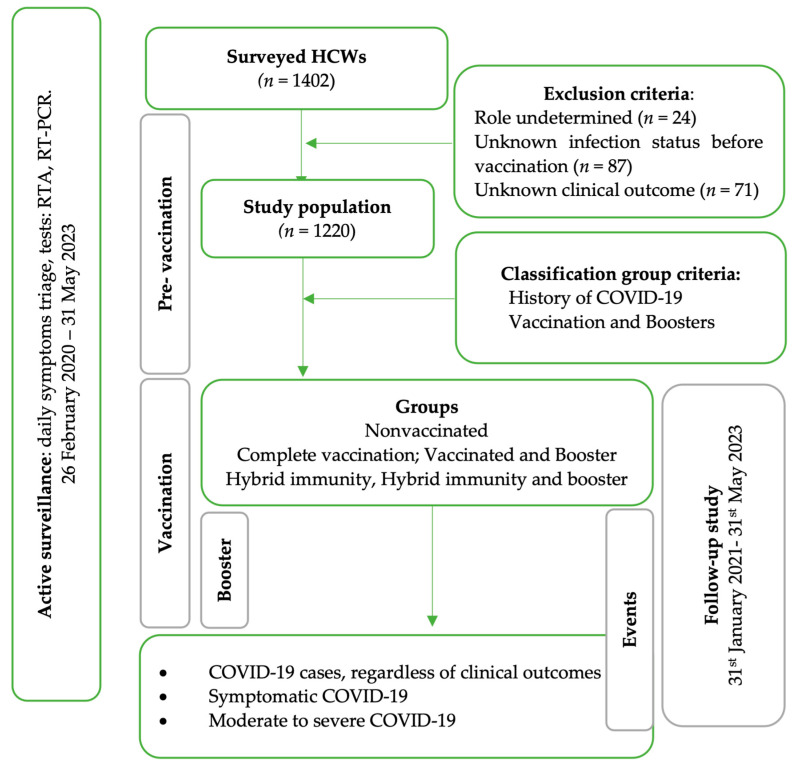
The workflow diagram for the active surveillance of HCWs in the National Institute for Infectious Diseases “Professor Dr. Matei Balș”, Bucharest, Romania, February 2020–May 2023.

**Figure 2 vaccines-12-00182-f002:**
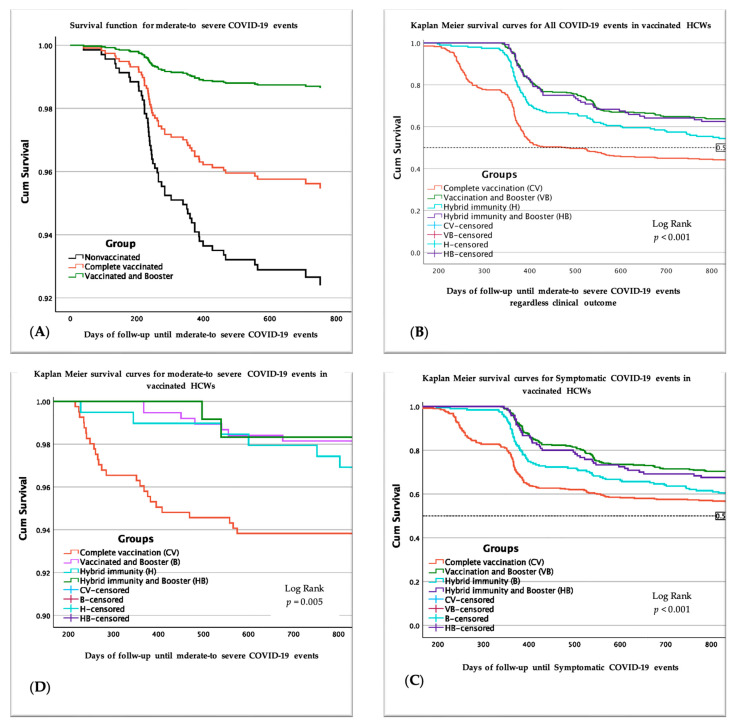
Survival curves related to COVID-19 events in HCWs followed up from 31 January 2021 to 31 May 2023, INBI ”Prof. Dr. Matei Balș”, Bucharest. (**A**) Survival curves of moderate to severe events in completely vaccinated (*n* = 600), vaccinated and booster (*n* = 498), and nonvaccinated HCWs (*n* = 122). (**B**) Kaplan–Meier survival curves related to all COVID-19 events in complete vaccinated (*n* = 405), vaccinated and boosters (*n* = 378), hybrid immunity (*n* = 195), hybrid immunity and booster (*n* = 120). (**C**) Kaplan–Meier survival curves related to symptomatic COVID-19 events in vaccinated HCWs. (**D**) Kaplan–Meier survival curves related to moderate to severe events in vaccinated HCWs. Log-rank test was used for comparison of equality of survival curves of vaccinated groups: complete vaccination, vaccinated and booster, hybrid immunity, hybrid immunity, and booster.

**Table 1 vaccines-12-00182-t001:** Characteristics of the study population (*n* = 1220) by laboratory evidence of SARS-CoV-2 infection, National Institute for Infectious Diseases “Professor Dr. Matei Balș,” Bucharest, Romania, February 2020–May 2023.

Characteristics	All HCWs	Laboratory Confirmed Cases		Uninfected	*p*-Value ^a^
	*n* = 1220 (%)	First Episode *n* = 767 (62.9 ^1^)	Second Episode *n* = 221 (18.1 ^2^)	*n* = 453 (37.1 ^3^)	
Gender					
Female	1020 (83.6)	645 (84.1)	186 (84.2)	375 (82.8)	0.5535
Male	200 (16.4)	122 (15.9)	35 (15.8)	78 (17.2)	
Age					
Mean (SD)	42.3 (10.4)	42.8 (9.9)	43.1 (9.2)	41.3 (11.1)	0.0147 ^b^
Median age (years); IQR	43.1; 34.2–50.4	43.5; 35.8–50.4	43.5; 36.3–50.5	41.7; 31.2–50.3	
≤29 years	181 (14.8)	91 (11.9)	21 (9.5)	90 (19.9)	0.0001
30–39 years	295 (24.2)	184 (24.0)	56 (25.3)	111 (24.5)	0.8438
40–49 years	410 (33.6)	278 (36.2)	82 (37.1)	132 (29.1)	0.0112
≥50 years	334 (27.4)	214 (27.9)	62 (28.1)	120 (26.5)	0.5964
Professional role					
Nurses	459 (37.6)	320 (41.7)	96 (43.4)	139 (30.7)	0.0001
Physicians	257 (21.1)	150 (19.6)	42 (19.0)	107 (23.6)	0.0981
Healthcare auxiliary staff	298 (24.4)	179 (23.3)	51 (23.1)	119 (26.3)	0.2388
Other roles	206 (16.9)	118 (15.4)	32 (14.5)	88 (19.4)	0.0717
Department					
High risk ^4^	844 (69.2)	547 (71.3)	170 (76.9)	297 (65.6)	0.0373
Low risk ^5^	376 (30.8)	220 (28.7)	51 (23.1)	156 (34.4)	

Abbreviations: IQR, interquartile range, SD standard deviation. ^a^ Calculated using the χ^2^ test using MedCalc Software Ltd. (Ostend, Belgium) comparison of proportions calculator, (https://www.medcalc.org/calc/comparison_of_means.php comparison of means calculator, Version 20.218; accessed on 25 October 2023). ^b^ Differences between the means in two independent samples using MedCalc Software Ltd. comparison of means calculator (Version 20.218; accessed on 25 October 2023). ^1^ Percent of total HCWs from the study (*n* = 1220), ^2^ Percent of total HCWs from the study, ^3^ Percent of total HCWs from the study, ^4^ intensive care units, emergency rooms, COVID-19 wards, ^5^ administrative laboratories, administrative, research departments.

**Table 2 vaccines-12-00182-t002:** Vaccination data of the study population (*n* = 1220) by laboratory evidence of SARS-CoV-2 infection, National Institute for Infectious Diseases “Professor Dr. Matei Balș”, Bucharest, Romania, February 2020–May 2023.

Characteristics	All HCWs	Laboratory Confirmed Cases		Uninfected	*p*-Value ^a^
	*n* = 1220 (%)	First Episode *n* = 767 (62.9)	Second Episode *n* = 221 (18.1)	n = 453 (37.1)	
Complete vaccination					
Complete vaccination (CV) *	1098 (90.0)	682 (89.0)	193 (87.3)	416 (91.8)	0.1033
CV before the first episode	783 (71.3)	367 (47.8)	51 (23.1)		
CV after the first episode	315 (28.7)	315 (41.1)	142 (64.3)		
Nonvaccinated	122 (10.0)	85 (11.1)	28 (12.7)	38 (8.2)	0.1033
Vaccine product					
Comirnaty	1031 (93.9)	638 (93.5)	183 (94.8)	393 (94.5)	0.4818
Jcovden	57 (5.2)	39 (5.7)	9 (4.7)	18 (4.3)	0.2866
Spikevax	9 (0.8)	5 (0.7)	1 (0.5)	4 (1.0)	0.5727
Vaxzevria	1 (0.1)	0 (0.0)	0 (0.0)	1 (0.2)	0.2155
Booster					
Booster (B)	524 (43.0)	283 (36.9)	72 (32.6)	241 (53.2)	<0.0001
B before the first episode	378 (31.0)	137 (17.9)	18 (8.1)		
B after the first episode	146 (12.0)	146 (19.0)	54 (24.4)		
B Homologous	516 (98.5)	281 (99.3)	71 (98.6)	235 (97.5)	0.0090
B Heterologous	8 (1.5)	2 (0.7)	1 (1.4)	6 (2.5)	

^a^ Calculated using the χ^2^ test using MedCalc Software Ltd. comparison of proportions calculator, (Version 20.218; accessed on 25 October 2023). * Healthcare workers vaccinated until the end of the surveillance period.

**Table 3 vaccines-12-00182-t003:** Characteristics of COVID-19 first and second episodes in HCWs, by clinical outcome and hospitalization, National Institute for Infectious Diseases “Prof. Dr. Matei Balș”, Bucharest, Romania, February 2020–May 2023.

Characteristics	First COVID-19 Episode		Second COVID-19 Episode		*p*-Value ^a^
	*n* = 767	%	*n* = 221	%	
Clinical outcomes					
Asymptomatic	144	18.8	25	11.3	0.0091
Mild	444	57.9	176	79.6	<0.0001
Moderate	159	20.7	20	9.0	<0.0001
Severe	20	2.6	0	0.0	0.0155
Hospitalization					
Yes (patient in ward)	310	40.4	39	17.6	<0.0001
No (outpatient)	457	59.6	182	82.4	
ICU	51	16.5 ^1^	2	5.1 ^1^	<0.0001
Mean LOS (days)	11 (SD 8.2)		6 (SD 5.0)		<0.0001 ^b^
Median LOS (days); IQR	9; 5–14		4; 3–7		

Abbreviations: ICU, intensive care unit; LOS, length of stay. ^a^ Calculated using the χ^2^ test using MedCalc Software Ltd. comparison of proportions calculator, https://www.medcalc.org/calc/comparison_of_proportions.php (Version 20.218; accessed on 25 October 2023). ^b^ Difference between the means in two independent samples using MedCalc Software Ltd. comparison of means calculator (https://www.medcalc.org/calc/comparison_of_means.php, Version 20.218; accessed on 25 October 2023. ^1^ Percent calculated from the hospitalized HCWs.

**Table 4 vaccines-12-00182-t004:** Cox proportional regression model for risk of all COVID-19 events, COVID-19 symptomatic events and moderate to severe COVID-19 events in HCWs, National Institute for Infectious Diseases “Prof. Dr. Matei Balș”, Bucharest, Romania, 31 January 2021–31 May 2023.

Covariate	All COVID-19 Events *		Symptomatic COVID-19 Events		Moderate to Severe COVID-19 Events	
	aHR (95%CI)	*p* Value	aHR (95%CI)	*p* Value	aHR (95%CI)	*p* Value
Age, gender						
<29 years	Reference	0.298	Reference	0.251	Reference	0.015
30–39 years	1.19 (0.89–1.58)	0.236	1.33 (0.97–1.83)	0.076	3.90 (0.86–17.37)	0.074
40–49 years	1.30 (0.99–1.72)	0.064	1.36 (1.00–1.87)	0.053	3.93 (0.88–17.47)	0.072
>50 years	1.15 (0.86–1.54)	0.338	1.30 (0.99–1.79)	0.114	7.89 (1.81–34.35)	0.006
Gender female	Reference	-	Reference	-	Reference	-
Gender male	0.93 (0.74–1.17)	0.540	0.87 (0.68–1.12)	0.284	0.51 (0.26–0.99)	0.046
Professional role						
Physicians	Reference	<0.001	Reference	0.002	Reference	0.174
Nurses	1.16 (0.85–1.58)	0.339	1.07 (0.77–1.49)	0.683	1.05 (0.40–2.75)	0.916
Healthcare auxiliary staff	1.14 (0.87–1.49)	0.356	0.94 (0.70–1.27)	0.705	0.76 (0.33–1.79)	0.534
Other roles	0.74 (0.55–1.01)	0.057	0.63 (0.45–0.89)	0.008	0.42 (0.15–1.79)	0.078
Working in high-risk department	1.29 (1.04–1.59)	0.020	1.44 (1.13–1.83)	0.003	2.95 (1.37–6.36)	0.006
Vaccination						
Nonvaccinated	Reference	<0.001	Reference	<0.001	Reference	<0.001
Completely vaccinated	1.10 (0.84–1.43)	0.507	0.81 (0.49–1.33)	0.396	0.59 (0.29–1.18)	0.134
Vaccinated and booster	0.71 (0.54–0.96)	0.023	0.23 (0.12–0.46)	<0.001	0.17 (0.07–0.43)	<0.001

Abbreviation: aHR, adjusted hazard ratio. * Laboratory confirmed COVID-19 events, regardless of clinical outcomes.

**Table 5 vaccines-12-00182-t005:** Cox proportional regression model for risk of all COVID-19 events, COVID-19 symptomatic events, and moderate to severe COVID-19 events in vaccinated HCWs, National Institute for Infectious Diseases “Prof. Dr. Matei Balș”, Bucharest, Romania, 31 January 2021–31 May 2023.

Covariate	All COVID-19 Events *		Symptomatic COVID-19 Events		Moderate to Severe COVID-19 Events	
	aHR (95%CI)	*p*-Value	aHR (95%CI)	*p*-Value	aHR (95%CI)	*p*-Value
Age, gender						
<29 years	Reference	0.076	Reference	0.218	Reference	0.014
30–39 years	1.30 (0.97–1.75)	0.08	1.40 (1.01–1.95)	0.045	7.10 (0.90–55.71)	0.062
40–49 years	1.47 (1.10–1.97)	0.010	1.37 (0.98–1.90)	0.064	8.52 (1.08–66.93)	0.045
>50 years	1.27 (0.93–1.72)	0.13	1.31 (0.93–1.84)	0.123	16.94 (2.20–130.73)	0.007
Gender female	Reference	-	Reference	-	Reference	-
Gender male	0.96 (0.76–1.23)	0.774	0.92 (0.70–1.19)	0.511	16.51 (2.14–127.44)	<0.001
Professional role						
Physicians	Reference	<0.001	Reference	0.005	Reference	0.301
Nurses	1.22 (0.89–1.69)	0.216	1.21 (0.86–1.71)	0.281	2.11 (0.60–7.42)	0.244
Healthcare auxiliary staff	1.10 (0.83–1.46)	0.521	0.98 (0.71–1.34)	0.891	1.26 (0.39–4.09)	0.704
Other roles	0.70 (0.50–0.96)	0.028	0.69 (0.49–0.98)	0.039	0.88 (0.25–3.11)	0.838
Working in high-risk department	1.87 (1.39–2.51)	0.001	1.87 (1.35–2.58)	<0.001	2.75 (1.08–7.04)	0.035
Vaccination and hybrid immunity						
Complete vaccination	Reference	<0.001	Reference	<0.001	Reference	<0.001
Vaccinated and booster	0.42 (0.33–0.51)	<0.001	0.48 (0.37–0.61)	<0.001	0.22 (0.09–0.52)	<0.001
Hybrid immunity	0.60 (0.47–0.77)	<0.001	0.76 (0.58–0.99)	0.041	0.40 (0.16–0.99)	0.047
Hybrid immunity and booster	0.42 (0.30–0.58)	<0.001	0.52 (0.36–0.74)	<0.001	0.15 (0.03–0.66)	0.012

Abbreviation: aHR, adjusted hazard ratio. * Laboratory confirmed COVID-19 events, regardless of clinical outcome.

## Data Availability

The datasets generated and analyzed during the current study are available from the corresponding author upon reasonable request.

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
