# Peer review of "The Dynamic Risk of COVID-19-Related Events in Vaccinated Healthcare Workers (HCWs) from a Tertiary Hospital in Bucharest, Romania: A Study Based on Active Surveillance Data"

_vaccines, 2024, doi:10.3390/vaccines12020182_

Round 1

Reviewer 1 Report

Comments and Suggestions for Authors

I am grateful to the editor for inviting me to review this intriguing paper on the evolving risk of COVID events among vaccinated healthcare workers (HCWs) in relation to their vaccination status.

The study unequivocally illustrates the efficacy of receiving a booster dose for high-risk operators. Additionally, adopting hybrid immunization demonstrated a reduced infection risk compared to a uniform immunization approach.

The duration elapsed since the vaccine administration is widely as a significant factor influencing COVID infection risk in vaccinated operators. This crucial variable, unfortunately, was not assessed in the statistical analysis. The trend of anti-S response in the vaccinated individuals might play a significant role in explaining the risk trends in the months following COVID vaccination. (DOI: 10.1093/pubmed/fdad198; 10.3390/vaccines10111967; 10.3390/vaccines9090947) . The BNT162b2 and the mRNA-1273 induce an IgA antibodies response, reflecting the possible prevention of the asymptomatic spread (10.1080/21645515.2022.2037384) Authors should in-depth address this limitation in the appropriate section.

The alarmingly low number of operators who have received a booster dose raises concerns. A more thorough discussion is warranted to explore the underlying reasons for this low vaccine uptake.

Reviewer 2 Report

Comments and Suggestions for Authors

This is a nice study design but the statistical analysis is naïve. It does not take into account repeated events nor does it adjust for potentially confounding variables. It must be corrected. Please consult an epidemiologist / biostatistician.

Please define 3 events more fully: any vs symptomatic vs moderate/severe

Define the “high-risk” departments

 MINOR ISSUES

LINE                      ISSUE

134-135              Two deaths….is a result, not a method. move to results section

Figures 2 & 3      Colour choices make it very hard to determine which group the lines represent and it will be even harder in smaller (i.e., published) figures

Comments on the Quality of English Language

 English needs to be polished throughout the manuscript

Examples

LINES                    ISSUE

40-42                   Sentence starting with “To this event…” is hard to follow

44                         Fix sentence

55                         The term “variants” is usually reserved for variants of a virus like SARS-CoV-2

Reviewer 3 Report

Comments and Suggestions for Authors

Summary:

The manuscript provides a long-term evaluation of the impacts of vaccination and infection in healthcare workers in a hospital in Romania. The authors investigate the impact and hazard ratios associated with symptoms, duration of illness, and severity of disease in multiple groups, including those that were fully vaccinated, fully vaccinated and boosted and hybrid immunity and booster groups. The results highlight the importance that HCWs are fully vaccinated and boosted to ensure improved protection against COVID-19 infection.

Major comments:

The study period is misleading and confusing to the reader. The title mentions that the study period was 3 years, while the abstract mentions 38 months of surveillance. However, the methods state that the study period was 28 months (line 113 in methods and line 279 in discussion). While the time of surveillance is factually different from the study period, I believe that the title and this mention of 38 months in the abstract is misleading to the reader. Please amend this to match the study period mentioned in the methodology (assuming that the information stated in the methodology is correct).

Minor comments:

Line 22: 38 months is stated here, while in the methods (line 113) it is stated as 28 months. Please adjust accordingly.

Line 44: This is unclear. I would suggest deleting the words “with o”

Line 48: For clarity, please provide the definition for “fully vaccinated” – this is listed in the methods, but should be included here in the introduction also.

Line 49: The use of the word “Protected” can be misleading as the people are not protected from the disease – as evidenced from the results. Therefore, consider an alternative word such as “vaccinated”.

Line 54: “and up to date” - Please provide the most recent date for this to the reader (e.g., January 2024).

Line 83: Please define “IPC team”.

Line 88-90: “Asymptomatic HCWs were…” please rewrite this sentence for clarity.

Line 100: The definition for the acronym “HCWs” is defined here, but the acronym appears much earlier in the manuscript.

Line 106 – 107: “we included only the HCWs with a maximum of two COVID-19 episodes.” – please explain as to why this was selected. Why were those that experienced more than 2 episodes excluded? What was the reasoning behind this decision?

Line 125 (and throughout manuscript): Please keep terminology consistent. Here “primary schedule” is used as opposed to “complete vaccination”. I would suggest using “complete vaccination” as this has been defined for the reader.

Line 167: Please write out the acronym IQR for clarity.

Table 1: I would suggest that the percentages of uninfected be adapted to the percentage of the represented population rather than a percentage of the entire population. For example, the percentage shown for HCWs under 29 years old in uninfected is the percentage of the total uninfected (90/453). Rather, to support the argument that the authors are making, it would likely be better to represent the percentage uninfected of the total population subset – therefore, for people under 29 years, it would be 90/181 (49,7%). This then depicts that a larger subset of a particular group remains uninfected.

Line 190: in the footnotes for the table caption, “emergency” should be “ER” to align with the abbreviations used in the same table.

Table 2: The two columns under the header “laboratory confirmed cases” should be labelled.

Lines 212-216: “in the immunologically naive HCWs group,  we  had 405 (46.28%) fully vaccinated subjects…” These sentences are not clear. Perhaps I am misinterpreting the information, but it seems that the authors are stating that vaccinated individuals are immunologically naïve? Please clarify or correct as appropriate.

Line 235: Spelling “an aHR”

Line 236: “events and 0.19 (95%CI: 0.08-0.46)…” Please clarify what the 0.19 HR relates to. It is assumedly of moderate and severe events

Table 5: Spelling “HR”

Line 249 – 251: “The hybrid immunity and booster group had the lowest risk for COVID-19-related events regardless of severity…” According to table 5, the Vaccine and booster group had a lower HR for COVID regardless of severity (0.46 compared with 0.48). Please amend accordingly. This must also be amended in the abstract (line 31-32) that states the same results. However, the vaccinated and booster groups seemingly performed better than the hybrid immunity and booster groups for 2 of the 3 risk categories mentioned.

Figures: Please consider changing the colors of the lines as it is difficult to differentiate the non-vaccinated and the vaccinated with booster lines – especially in the legend.

Comments on the Quality of English Language

The English language is suitable, however, I would recommend revision and editing as there are some sentences that are unclear to the reader and can cause confusion if not amended. 

Reviewer 4 Report

Comments and Suggestions for Authors

Thank you for sharing your article on risks of COVID-19-related events among vaccinated HCWs in a Romanian hospital setting. Here some comments that may help to improve the article:

L44: What's the meaning of "with o with"? Please revise.

L48: Presumably "the age group of 25-49" years? Please don't forget to add a unit. This applies throughout.

L54: "up to date" or up to today? Please revised as needed.

L56: Please specify the time point of "current" related to the vaccination schedules.

L59: Why were they considered at high risk only in the context of Omicron?

L78: Which criteria were applied to classify hospital departments as high-risk and low-risk? Why was this classification performed at all? 

L90: Were positive cases not separated from others, especially in a hospital setting? What measures were undertaken to prevent further viral spread through confirmed positive cases? 

L99: Which inclusions criteria? Please clarify. L106: Same here, which criterion? In case you mean eligibility criteria, please incorporate them clearly in your manuscript. 

L99-100: So, your cohort was not homogeneous, but heterogeneous due to constant in- and out-movements. Why have you decided to study such a cohort? What's your rationale?

L134: Is it death due to COVID-19 or possibly another underlying reason in addition to COVID-19?

L135-136: Please be more specific to potential readers what you mean by "validated manually and automatically" as well as "a data engineer's automation matched multiple sets". What do you mean by multiple sets? How is your database composed? 

L156: Did participant consent too? If so, please include informed consent procedures. 

Table 1: Please check the journal's guidelines how to report p-values, i.e., the number of decimal places. It seems that males were your reference group. What's the reference group for the variable age and job category? Please also check Table 2 and Table 3 for the same matter.

L340: Which specific biases are you referring to? Why not mentioning them in the manuscript? 

Comments on the Quality of English Language

Please see above. 

Reviewer 5 Report

Comments and Suggestions for Authors

The research is well presented, appears to have been conducted soundly, and provides interesting results.

The authors should consider a couple of points to help strengthen the manuscript.

You should provide clear evidence that assumptions underlying the Cox proportional model are satisfied.

The results seem to provide a good basis for specifying implicatiions for policy and practice. Adding a couple of paragraphs speaking to these points would greatly improve the opportunity for the research to have broader impact.

Comments on the Quality of English Language

Only minor editing for language seems necessary.

Round 2

Reviewer 2 Report

Comments and Suggestions for Authors

Thank you for the major revisions you have done with the paper.

However, your statistical analysis and interpretation thereof are incorrect.

1) Using survival analyses requires that you start the follow-up at comparable times. As such, you cannot start everyone at Jan 31, 2021 if their 'group' changes over time (you start them at Day 0, whatever that DATE might be). There is no possibility that you followed every participant in the booster groups, for example, starting Jan 31, 2021. 

2) Your interpretation of hazard ratios is incorrect. For example, line 274-275: the hybrid immunity group is actually compared with the complete vaccination group. The hybrid group does NOT have the highest risk.

To help readers follow the progression:

1) Figure 1 needs to delineate which participants are in which group --> regarding vaccination and infection.

2) I question your use of the complete vaccination group as the referent. Why not use the unvaccinated?

Comments on the Quality of English Language

Much improved. Still a few areas that would be improved with another review. However, this is moot in comparison with the statistical issues.

Reviewer 4 Report

Comments and Suggestions for Authors

Thank you for sharing the revised manuscript that has much improved and for addressing most of my comments. Please expand your constent statement. The way it still is stated in your manuscript simply does not suffice from an ethical point of view. 

Comments on the Quality of English Language

Please see above.

Round 3

Reviewer 2 Report

Comments and Suggestions for Authors

NONE